# Development of a Seamless Forecast for Solar Radiation Using ANAKLIM++

**Isabel Urbich** [1,*] **, Jörg Bendix** [2] **and Richard Müller** [1]

[1] Department for Research and Development, Deutscher Wetterdienst, Frankfurter Straße 135, 63067 Offenbach, Germany; richard.mueller@dwd.de
[2] Department of Geography, Philipps-Universität Marburg, Deutschhausstraße 12, 35032 Marburg, Germany; bendix@staff.uni-marburg.de
[*] Correspondence: isabel.urbich@dwd.de; Tel.: +49-(0)69-8062-2475

**Abstract:** A novel approach for a blending between nowcasting and numerical weather prediction (NWP) for the surface incoming shortwave radiation (SIS) for a forecast horizon of 1–5 h is presented in this study. The blending is performed with a software tool called ANAKLIM++ (Adjustment of Assimilation Software for the Reanalysis of Climate Data) which was originally designed for the efficient assimilation of two-dimensional data sets using a variational approach. A nowcasting for SIS was already presented and validated in earlier publications as seamless solar radiation forecast (SESORA). For our blending, two NWP models, namely the ICON (Icosahedral Non-hydrostatic model) from the German weather Service (DWD) and the IFS (Integrated Forecasting System) from the European Centre for Medium-Range Weather Forecasts (ECMWF), were used. The weights for the input data for ANAKLIM++ vary for every single forecast time and pixel, depending on the error growth of the nowcasting. The results look promising, since the root mean square error (RMSE) and mean absolute error (MAE) of the blending are smaller than the error measures of the nowcasting or NWP models, respectively.

**Keywords:** blending; nowcasting; numerical weather prediction; seamless prediction; solar radiation

## 1. Introduction

The contribution of photovoltaic (PV) power to the electricity mix increased significantly over the last decades in accordance with the overall importance of renewable energies [1,2]. Spatially and temporally high resolved solar radiation forecasts are strongly required due to the rapid installation of solar power worldwide and the integration of fluctuating PV power into the grid [3]. Therefore, it is essential for management and operation strategies that solar radiation forecasts deliver reliable predictions of the expected PV power for the next 0–12 h [1,4–7]. Because weather and power forecasts always contain a part of uncertainty there will be balancing costs for transmission system operators (TSO) related to forecast errors. However, these costs can be decreased by the reduction of the respective forecast errors [8]. An example from the Dutch electricity market shows that sophisticated forecasts are able to improve the average profit from electricity, which was generated by wind power plants. An advanced forecast model (24.7 EUR/MWh) is superior over persistence (22.4 EUR/MWh) by 10% and a perfect forecast (28.4 EUR/MWh) could lead to an increased income of 15% [9]. With regard to the rising share of renewable energies as a source of electricity, the need for more accurate forecasts is growing rapidly.

Usually, a nowcasting (NWC) covers the first few hours of forecast due to its higher temporal and spatial resolution [1,10–12]. However, observation based forecasts are known for their rapid error growth [3,13,14]. Therefore, numerical weather prediction (NWP) is used after the first couple

of hours, because of their ability to explicitly and systematically simulate atmospheric processes and their evolution at larger scales [15–17]. The point of interception, where NWP performs better than nowcasting, depends on numerous effects, like the weather situation, spatial and temporal resolution, data assimilation, time of initiation, and many more [18,19]. Thus, it is crucial to design a suitable blending of NWP and nowcasting to enable a seamless solar radiation forecast for transmission and distribution system operators as well as direct marketer. For renewable energies in general, blendings of different forecasts have been identified as a great possibility for improving the forecast quality for wind velocity as well as solar radiation [20–22]. It is not uncommon that combined products outperform all individual forecast products for the entire forecast horizon, since a blending takes advantage of their synergies and complementaries [10,20,23]. The approaches for blending multiple forecasts are numerous. A common established technique are artificial neural networks (NN) [24–30]. One reason for the popularity of NN is the opportunity to integrate PV energy yields into the forecast of PV power, without the need to simulate them, e.g., Wolff et al. [10]. Besides that, the resulting hybrid of several NWP models and satellite measurements has the ability to improve the quality of single forecasts of surface incoming shortwave radiation (SIS) as Marquez et al. [31] demonstrated. Other successful techniques for time series prediction approaches are fuzzy models [32], adaptive neuro-fuzzy interference systems [33], autoregressive models [34,35], multiplicative autoregressive moving-average statistical models [36], hidden Markov processes [37], multi-dimensional linear prediction filters [38], multi-model-mix of diverse forecasting approaches [39], and using the Weather Research and Forecasting model (WRF) to advect and diffuse Meteosat Second Generation (MSG) cloud index values [40]. Wolff et al. [10] compared a support vector regression with the linear regression of persistence, nowcasting, and NWP by Kühnert [41], and both of the methods performed very similarly. Besides that, like most of the blending approaches the combined forecast could outperform each individual forecast product. Kühnert [41] utilized a combination of the COSMO (Consortium for small-scale Modeling) model from the German Weather Service (DWD) and IFS (integrated forecasting system) from the ECMWF (European Centre for medium-range Weather Forecasts) for the NWP share in his regression. The mean RMSE (root mean square error) of the installed power for all stations in Germany was approximately 3% after 4 h and Kühnert [41] showed that the quality of the combined forecast can be significantly improved when the weighting parameters are selected day-time-dependent. A similar approach was presented by Lorenz et al. [1] for solar radiation, where they performed a linear regression of COSMO, IFS, and a nowcasting with the aim of developing a seamless transition between those forecasts. For a validation period of nine months, including nights, the RMSE was approximately 26 W/m$^2$ after a forecast horizon of 4 h. Haupt et al. [42] work on a blending system which performs a bias correction of the utilized models in a first step and after that optimizes the blending weights for each lead time according to their historical performance. This intelligent consensus forecast represents the NWP share which is later blended linearly with a nowcasting for the time period of 3–6 h [43,44]. The mean absolute error (MAE) of this hybrid method is dependent of the season and it ranges from 30 W/m$^2$ in January and February up to approximately 90 W/m$^2$ in May and June after 4 h respectively. Martinez et al. [45] are also using a linear transition for the combination of the nowcasting and the NWP model of the Spanish Weather Service between 0–4 h. After 4 h the relative RMSE for the global horizontal irradiance (GHI) equals 22% and 46% for the direct normal irradiance (DNI). With their optimum mix of satellite-derived cloud motion forecasts, the National Digital Forecast Database's (NDFD) cloud cover-derived irradiance forecasts and several operational numerical weather prediction models, Perez et al. [46] reported a relative RMSE of approximately 22% for GHI after a lead time of 5 h.

However, the methods that have been discussed so far are driven by optimization of blending techniques instead of meteorology. Thus, in this study we investigate a method which allows a blending depending on the regional weather situation aiming for a seamless temporal and spatial prediction. For this purpose, a novel approach for blending an satellite-based nowcasting with two NWP models, namely ICON (Icosahedral non-hydrostatic model) and IFS [47,48], while using a

software tool called ANAKLIM++ (Adjustment of Assimilation Software for the Reanalysis of Climate Data). It combines methods from data assimilation with Gaussian weights; thus, it can be extremely helpful for blending purposes in the scope of seamless forecasting. Originally ANAKLIM++ was designed for the efficient assimilation of two-dimensional data sets using a variational approach [49,50]. The aim of ANAKLIM++ was to assimilate NWP data, satellite data, and ground measurements in order to obtain climate data sets of SIS without data gaps. As for the nowcasting of SIS presented in Urbich et al. [51] the region of interest for the combined forecast is central Europe. ANAKLIM++ combines the nowcasting of SIS presented in Urbich et al. [51] with two NWP models for the forecast horizon of 1–5 h aiming for seamless temporal and spatial transition. The resulting integrated forecasts are validated with SARAH-2 (Surface Radiation Data Set–Heliosat) data from the Satellite Application Facility on Climate Monitoring (CM SAF). As error measures the bias, MAE and RMSE are calculated for a series of selected cases in August, September, and October 2017. These cases were selected due to their different weather situation over central Europe because clouds play a dominant role in solar radiation forecasts. Further, the performance of the blending with ANAKLIM++ is compared to a simple approach with a weighted mean.

In the following section, the utilized data which serves as input for the software tool ANAKLIM++ will be described. The data of both NWP models will be described in Section 2.1, while more information about the nowcasting can be found in Section 2.2. A description of the used reference data will follow in Section 2.3. The methodology of ANAKLIM++ (Section 3.1), a simple blending approach (Section 3.2), and the error measures used for the validation (Section 3.3) are explained in Section 3. After that, in Section 4, the results of the blending and the corresponding validation of the forecast are displayed and discussed.

## 2. Data

In the following, the data utilized for ANAKLIM++ are described in more detail. For a blending with ANAKLIM++ $N$ different two-dimensional data sets can be used as input. In order to be consistent with earlier publications [51,52], we examined a set of fifteen cases based on different weather situations for the months of August, September, and October 2017. A list of these cases can be found in the Appendix A in Table A1. The diversity of the selected weather situations in this study cover the relevant cloud types for PV forecasting in Central Europe.

### 2.1. Numerical Weather Prediction

For the blending with ANAKLIM++ two state of the art global NWP models, the ICON from the DWD, and the IFS from the ECMWF were selected.

ICON is the global and regional model of the DWD. The horizontal grid consists of a set of spherical triangles that seamlessly span the entire globe [47]. The horizontal resolution equals 13 km for the global and 6.5 km for the regional nest. The main runs are initialized four times a day at 00, 06, 12, and 18 UTC for the whole region up to 120 h and additional four times at 03, 09, 15, and 21 UTC with the EU (Europe) nest up to 30 h [47]. The runtime of ICON is approximately 3 h. SIS is given as accumulated solar radiation over several hours depending on the time of initiation in W/m$^2$. These accumulated values of SIS are recalculated as hourly averaged values to be similar to the other utilized forecasts. For this study, the global version of ICON is used, because the higher resolved ICON-EU did not lead to better validation results in particular compared to IFS [12,51]. Table 1 depicts a detailed list of all used forecasting products.

**Table 1.** Information on the forecasted solar radiation data used in this study.

| Product | Method | Operator | Horizontal Resolution | Temporal Resolution |
|---------|--------|----------|----------------------|---------------------|
| NWC | Short-term Forecast | DWD | 5 km | 15 min |
| ICON | Numerical Weather Prediction | DWD | 13 km | 1 h |
| IFS | Numerical Weather Prediction | ECMWF | 9 km | 1 h |
| SESORA | Blended Forecast | DWD | 5 km | 1 h |

IFS is the global NWP model of the European Center for Medium Weather Forecast (ECMWF). The deterministic run of the IFS comes with a horizontal resolution of 9 km and it is performed twice daily with initial times 00 and 12 UTC [48]. The runtime of the IFS is approximately 6 h. SIS is given in hourly accumulated values in $J/m^2$. However, for this study, all of the results were recalculated into hourly averages in $W/m^2$ to be comparable with ICON and the nowcasting.

For this study, only those model runs are used that would be actually available in an operational service. Hence, the different runtimes of IFS (6 h) and ICON (3 h) were considered for the selection of the forecast. A complete ICON run is available every 3 h whereas IFS runs are available at 06 and 18 UTC. Thus, for this study, the 06 UTC run from ICON and the 00 UTC run from IFS are taken, since the blending of all example cases will always start at 10 UTC.

Because the horizontal resolution of the NWP models does not equal the resolution of the nowcasting, both NWP models are interpolated into spacial grids of 0.05° before blending all data sets as it was also done by Mathiesen et al. [53]. The interpolation is performed with a nearest-neighbor-method. By doing this, we take the maximum advantage out of the higher resolved nowcasting.

*2.2. Nowcasting*

The SIS nowcasting used for this study has already been presented and validated in two previous publications as SESORA (seamless solar radiation forecast) [51,52]. It is based on the optical flow of the effective cloud albedo derived from the visible channel of MSG. The utilized method TV-$L^1$ from the open source library OpenCV [54] (Open Source Computer Vision Library) uses a multi-scale approach in order to capture cloud motions on various spatial scales. After the clouds are displaced SIS is being calculated by SPECMAGIC NOW (Spectrally-resolved mesoscale atmospheric global Irradiance Code for Nowcasting). This algorithm by Müller et al. [55] calculates global irradiation spectrally resolved from satellite imagery. As the effective cloud albedo, the nowcasting of SIS is available every 15 min. with a horizontal resolution of 0.05°.

As a preparation for the blending with ANAKLIM++, the hourly means were calculated. A very specific feature of the optical flow is the inward moving edge upstream of the cloud movement. Due to the fact that the method works without any kind of boundary conditions, the area behind the displacement contains no data. Like all pixels without data, this area is marked in white (Figure 2c,d). However, this area can be filled again with values in the process of the blending with ANAKLIM++.

*2.3. Reference Data*

For the validation of all forecasts used in this study, data from the CM SAF were used. Their SIS data from the SARAH-2 data set is the latest CM SAF climate data record of surface radiation based on the geostationary Meteosat satellite series [56]. The area covered by SARAH-2 spans from −65° to +65° in latitude and longitude with a horizontal resolution of 0.05°. The high quality of SARAH-2 in reference to the Baseline Surface Radiation Network [57] data is documented in Pfeifroth et al. [58].

**3. Method**

In this section, a brief overview of the blending method ANAKLIM++ and of the configurations utilized for this study is given. Moreover, the method of validation is described in this section.

### 3.1. ANAKLIM++

The aim of ANAKLIM++ is to assimilate multiple two-dimensional data sets to obtain a result that is closest to reality and has no missing data [50]. A general solution for a combination of $N$ two-dimensional data fields e.g., weather forecasts is to build the mean of these forecasts. A more sophisticated combination can be obtained by calculating a weighted mean

$$z(x) = \frac{\sum_l^N G_l(x) p_l(x)}{\sum_l^N G_l(x)} \tag{1}$$

with a real-valued function $p_l(x)$ and a weighting function $G_l(x)$ where $l \in [0, N-1]$ [49]. To find a new data set $z(x)$, which is similar to the input data $p_l(x)$, the minimum of

$$L(z(x)) = \sum_{l=0}^{N-1} G_l(x)(p_l(x) - z(x))^2 \tag{2}$$

has to be found. This further leads to solving the following equation:

$$0 \stackrel{!}{=} \frac{\partial}{\partial z(x)} L(z(x)) = \sum_{l=0}^{N-1} G_l(x) p_l(x) - z(x) \sum_{l=0}^{N-1} G_l(x). \tag{3}$$

This condition can be expanded with an additional term to penalize artifacts and large gradients and yielding smoother results. With $\lambda$ being a Lagrangian multiplier, this finally leads to

$$L(z(x), \nabla z(x)) = \sum_{l=0}^{N-1} G_l(x)(z(x) - p_l(x))^2 + \lambda \|\nabla z(x)\|^2. \tag{4}$$

By using the Euler–Lagrange-formalism, the minimization of the above can be transformed into a sparse system of linear equations, which can further be solved by a conjugate gradient solver [50].

As stated in Section 2.1, only those SIS forecasts are used for a blending, which would also be available for the operational service. For a better overview over the available data, a scheme of the temporal procedure can be found in Figure 1 and two exemplary cases are shown in Table 2. The required input forecasts for ANAKLIM++ are the hourly averaged SIS of ICON, IFS, nowcasting, and a reference, respectively. Because the reference is the analysis of our nowcasting both are available every 15 min. They are being recalculated into hourly averaged values for the blending. Thus, the blending will be done after 1 h of nowcasting. The temporal resolution of both NWP models is 1 h. The lead time of them depends on the initiation time and the runtime (Section 2.1).

**Table 2.** Two exemplary cases (08 and 09 UTC) for the temporal availability of the utilized forecasts for a blending. Depending on the start of the blending the lead time of the forecasts can differ.

| Product | Start of Blending | Runtime | Model Initialization | Time of Availability | Lead Time at Start of Blending |
|---------|-------------------|---------|----------------------|----------------------|--------------------------------|
| Nowcasting | 08 UTC<br>09 UTC | 15 min | 07 UTC<br>08 UTC | 07:15 UTC<br>08:15 UTC | 1 h<br>1 h |
| ICON | 08 UTC<br>09 UTC | 3 h | 03 UTC<br>06 UTC | 06:00 UTC<br>09:00 UTC | 5 h<br>3 h |
| IFS | 08 UTC<br>09 UTC | 6 h | 00 UTC<br>00 UTC | 06:00 UTC<br>06:00 UTC | 8 h<br>9 h |

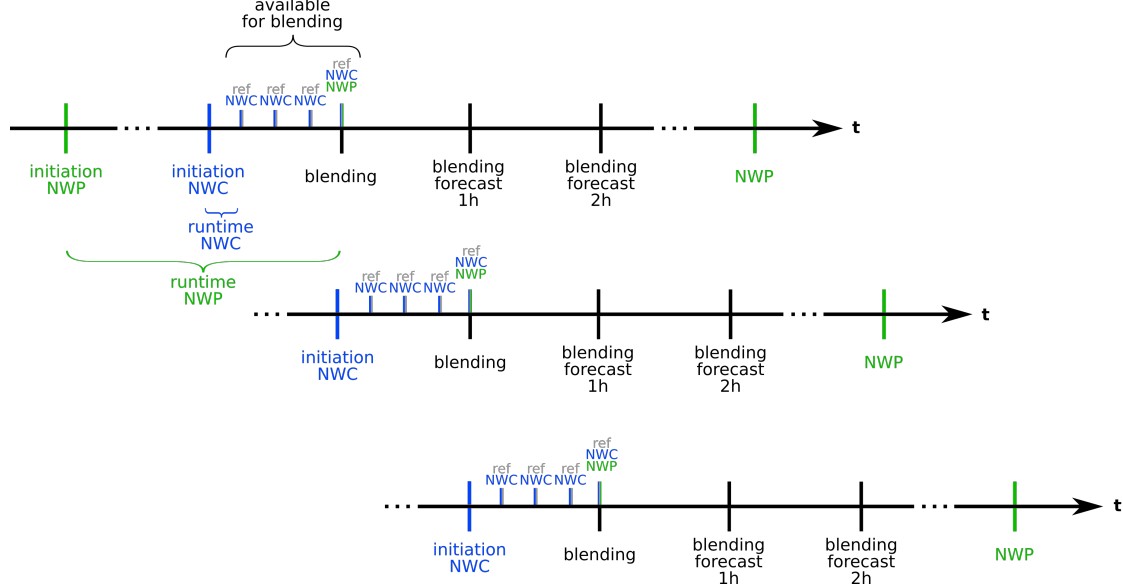

**Figure 1.** Scheme of the temporal availability of the input forecasts for a blending with ANAKLIM++ (adjustment of assimilation software for the reanalysis of climate data). The distance of the depicted forecasts is proportional to the time range in which they are available. The dots on the line mark a break in the time line due to a changing temporal availability and differing runtimes of the utilized NWP (numerical weather prediction) models. For each blending 1 h of NWP, NWC nowcasting and reference needs to be available. The procedure can be repeated every hour.

A simple blending with a weighted mean can only use the same weighting for each pixel in the data. A more sophisticated meteorological approach should be able to consider local weighting dependent on the regional weather, which, in turn, defines the quality of the forecasts. In order to meet this issue a key feature of ANAKLIM++ can be used. ANAKLIM++ was originally intended to fill gaps of missing satellite data. To make use of this feature, regions with rapid error growth can be detected in the analysis and the respective regions can be cut out of the nowcasting if the errors exceed a certain threshold. To locate those regions, the analysis of the nowcasting; hence, the 0-min.-forecast is compared to the 60-min. nowcasting and the absolute difference between these data fields is calculated. For the first blending all values over the 90th percentile of the absolute difference are cut out and filled by ANAKLIM++ with data from ICON and IFS. For further blending forecasts the percentile becomes lower and the cropped area becomes larger. In this manner, the weight of the nowcasting is depressed for e.g., convective weather situations which cannot be captured by pure advection of TV-$L^1$, but still used with full weight for weather situation characterized by pre-dominantly advective cloud movements. For these advective weather situations, a nowcasting outperforms NWP for several hours. See Urbich et al. [51] for further discussions on the relation between weather and nowcasting errors. This approach is used to allow a conservative forecast of the cut out areas, since we will never exactly know where errors of the nowcasting will occur. So far, we know that especially regions with forming or dissolving clouds e.g., convection cause problems to the optical flow method [51,52]. However, to not just ignore the cut out values they are utilized in an additional lower weighted layer in ANAKLIM++ from here on called an inverse layer. This way, the potential errors will be penalized, but valuable pixels can still have their impact on the resulting product. In the remaining area, ANAKLIM++ builds a weighted mean between all possible data sets, thus nowcasting, IFS, and ICON.

Under all possible configurations in ANAKLIM++, we think that the Lagrangian weights are the most practicable adjusting screw for our purpose. ANAKLIM++ provides the opportunity to choose Lagrangian weights for the similarity operator itself and for its first and second derivative [49]. Thus, only weights for the similarity operator itself were chosen and varied for each forecast time. A list of the chosen Lagrangian weights for ANAKLIM++ can be found in Table 3. The blending

starts with a large share of nowcasting and fades out slowly while the shares of IFS and ICON slowly increase. The inverse layer of the nowcasting with gaps, thus only the content that was cut out is used. However, the inverse layer is utilized with lower weights so that the sum of the Lagrangian weights of both layers slowly decreases. As a result, areas where errors are more likely to rise are penalized but the information of the troubled areas can be still used. All of the weights were particularly chosen as a consequence of the respective performance of the forecasts in comparison to the reference data. The results of this validation are depicted in Figure 4. Due to a better performance of IFS with regard to the RMSE and bias of SIS, the weights for IFS are higher than those of ICON throughout the entire blending period.

**Table 3.** Lagrangian weights for the blending with ANAKLIM++ for the nowcasting with gaps and its corresponding inverse layer (only content of gaps) and for the NWP models IFS (integrated forecasting system) and ICON (icosahedral non-hydrostatic model) respectively.

| Product | Lead Time (h) | | | | |
|---|---|---|---|---|---|
| | **1** | **2** | **3** | **4** | **5** |
| SIS NWC with gaps | 0.75 | 0.55 | 0.35 | 0.15 | 0 |
| SIS NWC inverse | 0.05 | 0.10 | 0.15 | 0.20 | 0.25 |
| IFS | 0.15 | 0.25 | 0.35 | 0.45 | 0.50 |
| ICON | 0.05 | 0.10 | 0.15 | 0.20 | 0.25 |

### 3.2. Simple Blending

To be able to evaluate the quality of a blending of NWP and nowcasting by the means of ANAKLIM++, a simple blending is performed for comparison. For this purpose, the original nowcasting without holes and the IFS is used. With those two data sets, a weighted mean of SIS is built for up to 4 h of forecast time. For the first hour, the nowcasting gets weighted with 80% and the weight decreases further, until it reaches 0%, while, for IFS, it is the other way around.

### 3.3. Error Measures

SIS of all forecast products was verified in the same manner. In a first step, the absolute difference between the forecast and SARAH-2 was determined. After that, the bias

$$\text{bias} = \frac{1}{n} \sum_{i=1}^{n} (x_i - y_i), \qquad (5)$$

mean absolute error (MAE)

$$\text{MAE} = \frac{1}{n} \sum_{i=1}^{n} |x_i - y_i|, \qquad (6)$$

and root mean square error (RMSE)

$$\text{RMSE} = \sqrt{\frac{1}{n} \sum_{i=1}^{n} (x_i - y_i)^2} \qquad (7)$$

were calculated. We decided to work with these error metrics, since they are used most commonly in the scope of energy meteorology and solar radiation forecasting [59]. The spread of the mean error metrics of all investigated cases is captured by means of the empirical or sample variance $v$

$$v = \frac{s^2}{\bar{x}}. \qquad (8)$$

Here, $\bar{x}$ is the mean

$$\bar{x} = \frac{1}{n} \sum_{i=1}^{n} x_i \tag{9}$$

and $s$ represents the standard deviation

$$s = \sqrt{\frac{1}{n-1} \sum_{i=1}^{n} (x_i - \bar{x})^2} \tag{10}$$

of $x_i$.

## 4. Results

Fifteen cases with different weather situations were examined for the months of August, September and October 2017 (Section 2). For the sake of clarity, the effect of the blending on the SIS structures is discussed based on two vivid examples in Figures 2 and 3. The overall results, showing the average of the bias, MAE, and RMSE for all cases will be presented in Figure 4.

Figure 2 shows the hourly averaged SIS for 7 August 2017 11 UTC. In Figure 2a the result from the ANAKLIM++ blending with a lead time of 60 min. is presented and Figure 2b shows the respective result of the reference data SARAH-2. An example of the hourly averaged nowcasting with and without data gaps where regions with high potential of significant error growths were detected is shown in Figure 2c,d. Figure 2e,f depict the results from the NWP models IFS and ICON.

The resulting blending in Figure 2a still shows the large structures from the nowcasting with a slight adaption of the amplitude of SIS. In this case, ANAKLIM++ led to a better forecast of SIS according to the reference data. Further changes also appeared in the area around the Mediterranean sea where the nowcasting shows quite high values of SIS ($>900\ \mathrm{W/m^2}$) in the forecast. Especially, IFS has lower values of SIS throughout a larger region. A blending that is more or less a weighted mean therefore leads to a medium result which for this example lies closer to the reference data as well. A further advantage of ANAKLIM++ is that it refills the missing data of the inward moving edge in our nowcasting with appropriate data of the NWP models.

Another example of a blending of NWP and nowcasting utilizing ANAKLIM++ is presented in Figure 3 for 30 September 2017 11 UTC. The results of the blending are depicted in Figure 3a with a lead time of 60 min. while Figure 3b shows the solar radiation of the reference data from SARAH-2. The corresponding input data (nowcasting, IFS and ICON) can be seen in Figure 3c–e.

A very similar behavior as for the other case can be recognized in the example of 30 September 2017 in Figure 3a. The overall structure of the frontal region looks very much like the structure of the front in the SIS nowcasting. Both NWP models predict a smaller shape of the front. Nevertheless, the SARAH-2 data in Figure 3b show a frontal structure, just like the one of the blended forecast. Here, the outer structure of the front is taken from the nowcasting while the inner structure is a result of all the input data. Over the North Sea behind this front, there is a small region, where SARAH-2 shows optically thinner clouds. This feature cannot be recognized in the nowcasting, since this area shows thicker clouds; however it is visible in the blending. This seems to be a result of the impact of ICON and IFS.

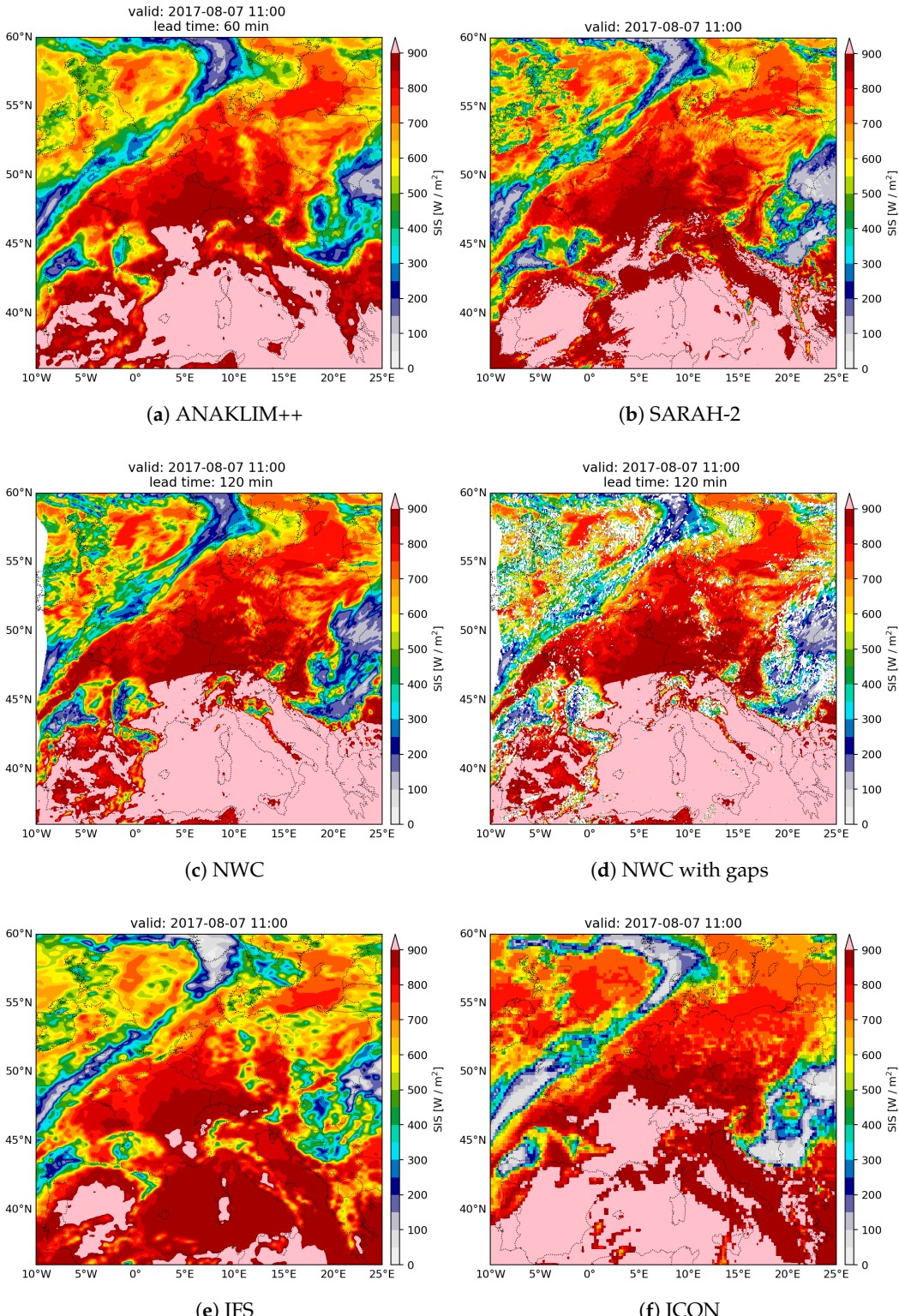

**Figure 2.** Hourly averaged SIS (surface incoming shortwave radiation) of (**a**) the blending with ANAKLIM++, (**b**) SARAH-2 (surface radiation data set–Heliosat), (**c**) NWC, (**d**) NWC with data gaps, (**e**) IFS and (**f**) ICON for 2017-08-07 11 UTC, respectively.

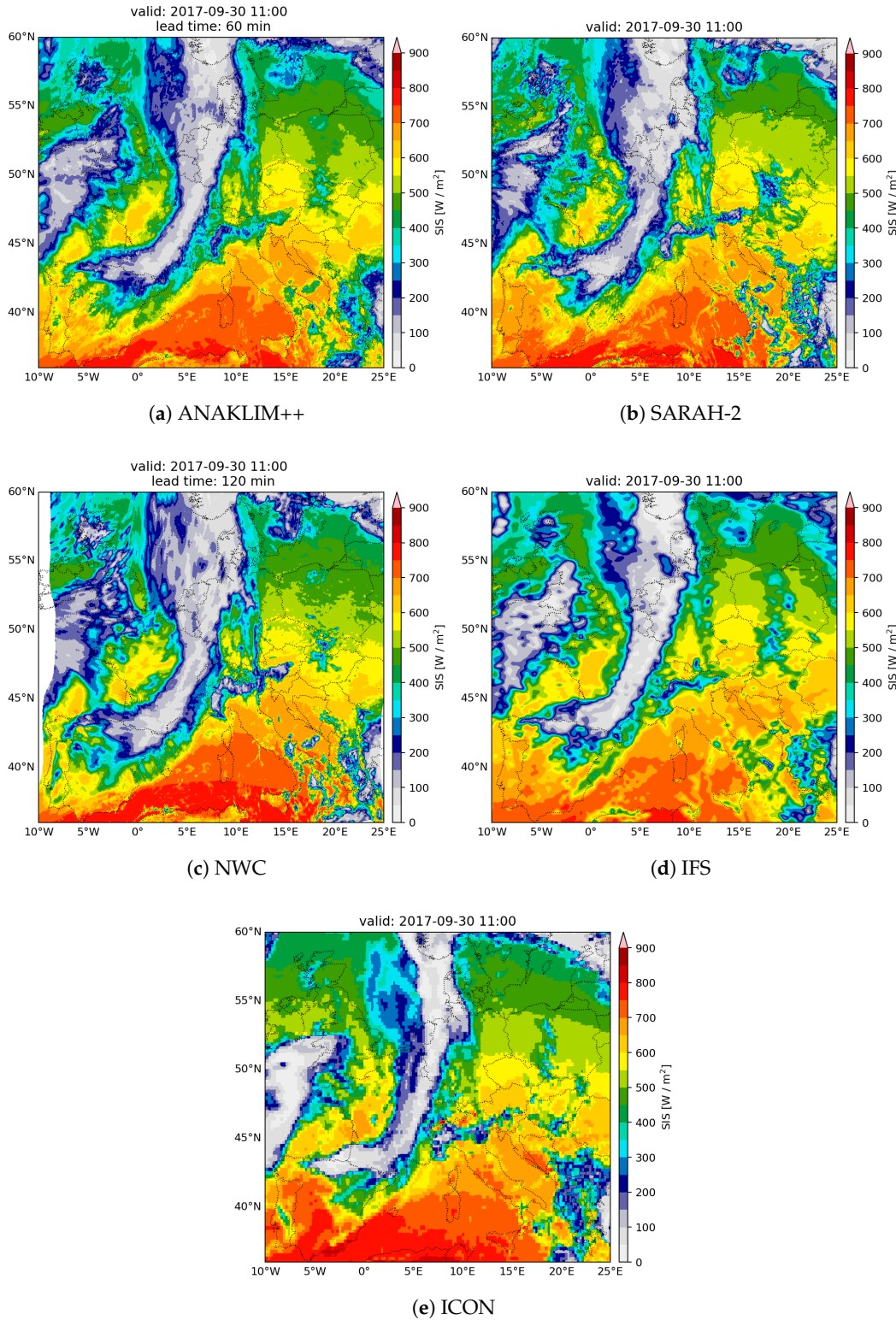

**Figure 3.** Hourly averaged SIS of (**a**) the blending with ANAKLIM++, (**b**) SARAH-2, (**c**) NWC, (**d**) IFS and (**e**) ICON for 2017-09-30 11 UTC, respectively.

After discussing the effect of the ANAKLIM++ blending on the SIS structures based on two illustrative cases the validation results for all examined cases will be discussed. Further, the ANAKLIM++ blending will be contrasted with a simple blending approach (Section 3.2), with persistence, with the

nowcasting, and with ICON and IFS. Figure 4 displays the results of this validation. For the persistence used in this validation, the position of the sun was corrected.

Figure 4b,c show the mean RMSE and mean absolute bias of all examined cases in dependency of the time of day. However, the time of day does not equal the lead time of these forecasts. As already mentioned in Section 2.1 only those model runs were used that would be available for a blending at 10 UTC; therefore, the 06 UTC run from ICON and the 00 UTC run from IFS are used. Moreover, the nowcasting and the persistence started at 09 UTC because one hour of SIS is needed as input. For better understanding the reader may want to look at the scheme of the temporal availability of the different data sets in Figure 1. Figure 4b,c show the error measures for a version of the blending where the nowcasting with data gaps is used. Both look quite similar because the RMSE and MAE of the blending with ANAKLIM++ are the lowest of all forecasts throughout the entire validation horizon. It is also similar in both figures that ICON shows the highest errors at the beginning and towards the end of the blending the persistence forecast has the highest errors. Moreover, the performance of both NWP models is slightly improving with forecast time, while the errors of the nowcasting and persistence slowly increase. A little surprise is that a simple blending approach can outperform the other forecasts. After a forecast time of 4 h, the difference between ANAKLIM++ and the simple approach becomes slightly larger, but all in all both blendings improved the quality of each forecast before the blending. Another similar behavior is that both error measures show a decrease for all forecasts towards longer lead times which is owed to the sunset. In Urbich et al. [51], it was already proved that this behavior is related to the sunset, since relative error measures did not decrease in the same manner. Figure 4a also depicts the RMSE of all examined cases. In addition, this figure shows the results of a blending utilizing ANAKLIM++, where the nowcasting data was used completely therefore no regions were cut out, see Figure 2c in comparison to Figure 2d for further details. This comparison shows that the approach of error filtering leads to a significant improvement (lower RMSE) after approximately 2 h forecast time when errors in the nowcasting get bigger and larger parts are being cut out in the version with data gaps. Simultaneously, the quality of the NWP models increases especially in comparison to the nowcasting. Therefore a blending with gaps in the nowcasting data in regions with high potential of errors can lead to lower error measures. In order to assess the quality of forecasts in the scope of energy meteorology it is crucial to consider the error spread as well. A low error can be misleading that the overall forecast quality is high for all weather situations. However, a misinterpretation of the forecast due to lack of information can affect grid stability. The error bars in Figure 4a–d depict the empirical variance $v$ (Equation (8)) thus it represents the spread of the error measures of the investigated cases. Because the simple blending only utilizes IFS for a lead time of 5 h both forecasts will always have the same size of error and variance. All forecasts show a rising spread with increasing lead time. Further, the spread of ANAKLIM++ is smaller than that of the simple blending for all presented error metrics.

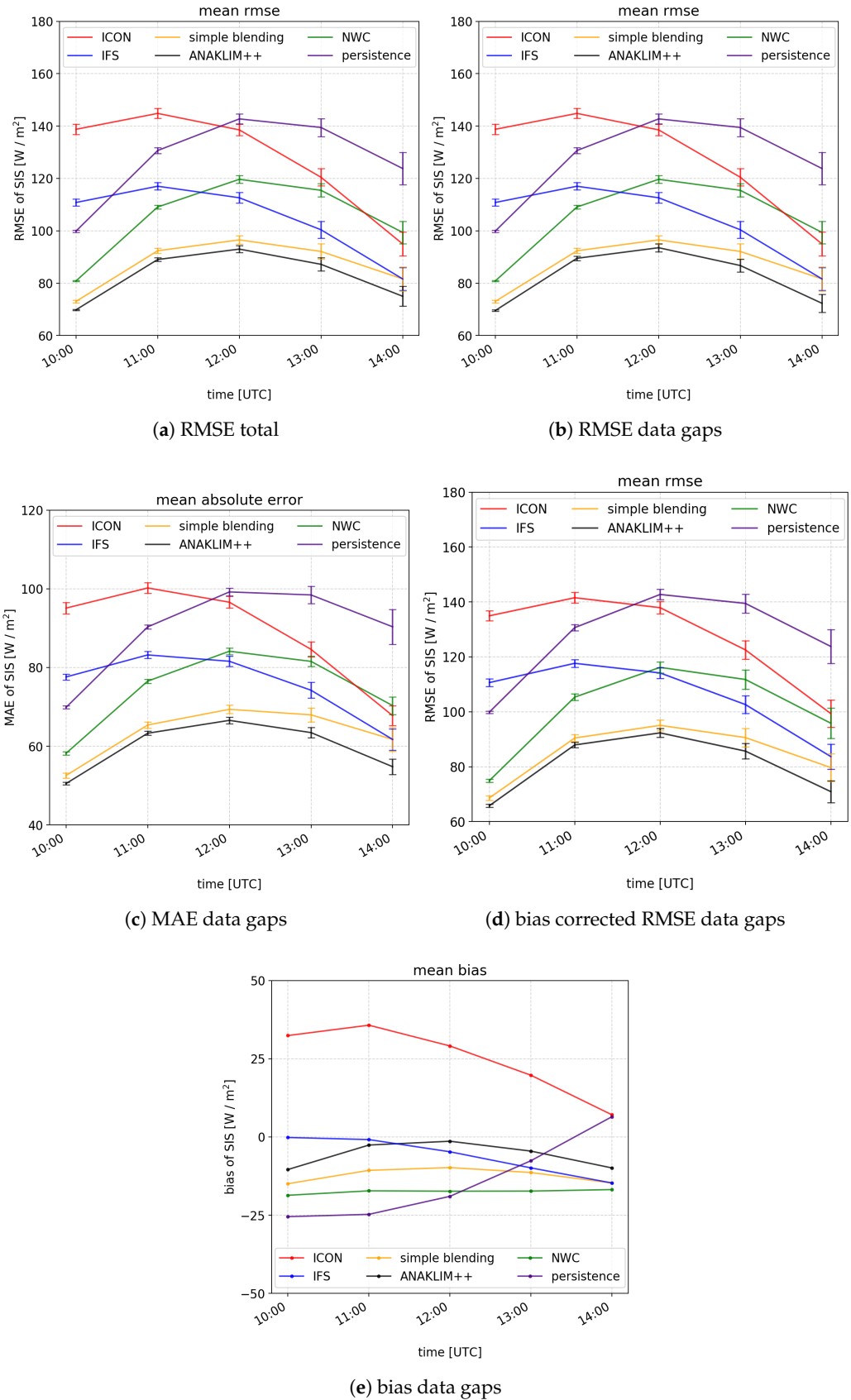

**Figure 4.** Mean error measures of all cases against day time for (**a**) RMSE (root mean square error) with whole NWC, (**b**) RMSE with gaps in NWC, (**c**) MAE (mean absolute error) with gaps, (**d**) bias corrected RMSE with gaps and (**e**) bias with gaps. The empirical variance is depicted by error bars for (**a**–**d**). The validation of SIS was performed with SARAH-2 data and the region of interest is Europe.

Figure 4e presents another error measure. Here, the mean bias of all cases is plotted against time for the version with gaps. The bias of the shown SIS forecasts reveals, among other things, why both blendings which where analyzed in this study performed better than all other forecasts individually. After calculating the weighted mean in both blendings the chosen input data has such an impact on the results that both lie closer to the reference data SARAH-2. Due to an overestimation of SIS in ICON and an underestimation of SIS in IFS and the nowcasting even a simple blending of those three forecasts can improve the quality of the blended forecast from the first hour. This can be confirmed by Figure 4d. In this figure, the bias corrected RMSE is displayed for the case of a nowcasting with data gaps. Because the input of the blendings was bias corrected, the difference between the RMSE of the nowcasting and the blended forecasts decreases significantly in comparison to Figure 4b, where no bias correction was applied. However, the RMSE of the blended products is still lower than the one of nowcasting, because the error cancellation of large positive values versus large negative values within the merging process is reduced, but not eliminated by the bias correction. As a consequence of this error cancellation, the blended forecasts show the lowest RMSE. Furthermore, the respective errors of ICON, IFS and nowcasting are lower as well after a bias correction. Figure 4e can also deliver a deeper understanding of the individual shares of each forecast for the blending method. For the nowcasting a bias was found which was already discussed in Urbich et al. [51]. Together with the knowledge that the RMSE of the nowcasting grows with increasing forecast time (Figure 4b), it seems reasonable to reduce the share of the nowcasting with time. Moreover, it appears that, for a successful blending, the share of IFS has to be larger than the one of the ICON model and that both shares should increase with time. Further, a smaller bias was found for the ANAKLIM++ blending than for the simple blending for each lead time. For the mean bias of the investigated cases the empirical variance is inadequate since the absolute values of the bias are much smaller, which leads to larger results of the variance (Equation (8)).

## 5. Discussion

The results that are presented in this study are consistent with those found in the literature. In this study a blending method was presented which uses a software tool called ANAKLIM++. For the ANAKLIM++ blending a RMSE of 72 W/m$^2$ and MAE of 55 W/m$^2$ after 4 h was found and the maximum is reached after 2 h, respectively. For a lead time of 2 h the RMSE equals 94 W/m$^2$ and the MAE is 67 W/m$^2$. After a forecast horizon of 4 h, the ANAKLIM++ blending shows an improvement of 13% regarding the RMSE and 11% regarding the MAE in comparison to the simple blending. Haupt et al. [42] report the maximum value of MAE for September and October after 7 h when the initialization lies between 09 and 12 UTC. In that case the MAE is approximately 70–75 W/m$^2$. All in all, the maximum MAE ranges between 60–125 W/m$^2$ for the whole validation period in 2015. Because of different regions of interest, error measures, validation periods, or initialization times, a comparison with other studies is complicated. Nevertheless, the results of this study show that ANAKLIM++ performs well as a blending tool in order to obtain a seamless prediction of SIS. The errors are in the same order of magnitude as those of Haupt et al. [42], Lorenz et al. [1], Martinez et al. [45] or Perez et al. [46]. Furthermore, the weight of the nowcasting can be regionally defined while taking into account errors that are induced by different weather situations. This is a strong feature, but also a challenge for a seamless prediction since large regional differences in weights are a source for breaks and inhomogeneities. However, the spatial and temporal development of structures and patterns of SIS show no break. This in turn is an essential basis for a seamless prediction of solar radiation. Further, the concept enables to take full benefit from improvements in the nowcasting method as well as NWP methods. This results in an enhancement of the quality of the seamless forecast up to 13% and 31% as compared to IFS and ICON and 37% in contrast to the nowcasting after 4 h.

Because of the fact that regions with fast growing errors are cut out the overall look of the blended forecast shows large structures from the nowcasting while overestimated and underestimated values of SIS are corrected towards a medium state. Figure 4e depicts the bias of all forecast products for

the time period of the blending. It becomes clear that a combined forecast of nowcasting, IFS, and ICON has to result in medium state, which, in this case, lies closer to the reference. As was already discussed in Urbich et al. [51], SPECMAGIC NOW which is used for calculating SIS produces a bias of about $-25$ W/m$^2$ for all lead times. This finding motivated the development of an improved version of SPECMAGIC NOW. ICON seems to overestimate SIS, which is a common fact for most NWP models ,while IFS only shows a small underestimation when compared to SARAH-2 [3,60]. The quality of each forecast can be better assessed by regarding the MAE or RMSE, which can be found in Figure 4b,c. Both of the plots show a similar behavior of the results. ICON seems to be troubling a bit more with the prediction of SIS than IFS. Moreover, it is a proven fact that IFS from the ECMWF is a high quality forecast which usually performs better than other NWP models concerning SIS [46,61,62]. The ANAKLIM++ blending, as well as the simple approach, deliver better results than each individual forecast product for up to 4 h. Especially after 13 UTC, the blending with ANAKLIM++ might have a small advantage over the simple approach. The RMSE and MAE of all forecasts decrease towards the evening because of the sunset. This effect can be canceled out by calculating relative error measures instead of absolute ones, as can be seen in Urbich et al. [51]. The relative error measures are, of course, important in evaluating the quality of forecasts; however, for the end users, an absolute error is simply more useful. The users of the seamless SIS forecast are transmission and distribution system operators as well as direct marketers, and they may use this forecast for trading and grid security. In this case, absolute errors are more useful, since they are more direct.

## 6. Conclusions

In this study, a novel approach for a blending of a nowcasting and two NWP models, namely ICON from the DWD and IFS from the ECMWF, was presented. The aim of this combination is a seamless forecast of SIS for a forecast horizon of 0–12 h. Further, this forecast should benefit from the higher temporal and spatial resolution from the nowcasting, as well as from the reliable prediction of NWP on larger time scales. The blending of these forecast products is done between 1–5 h with a software tool called ANAKLIM++. Its original purpose was the efficient assimilation of two-dimensional data sets using a variational approach [50]. In this study ANAKLIM++ has been optimized and applied to a blending of NWP and nowcasting. The nowcasting of solar radiation is based on the optical flow of the effective cloud albedo derived from the visible channel of MSG [51,52]. Furthermore, we applied a simple blending as a benchmark for ANAKLIM++. Both blendings and all input data were validated with SARAH-2 data from the CM SAF and further the bias, MAE and RMSE were calculated for each lead time, respectively. In this study, 15 cases were examined from August, September, and October 2017 due to their different weather situation and cloud patterns over central Europe (Table A1). A special feature of ANAKLIM++ is that it was designed to fill data gaps for assimilation purposes. Thus, we took advantage of this feature and decided to cut out areas of potential error growth in the nowcasting to fill it with NWP data instead. The affected areas are still used for the blending but with a lower weight. That way, the areas of potential error growth are being penalized, but they are still able to deliver additional information to the blending. The results show that ANAKLIM++ is well suited for the seamless prediction of solar radiation. The spatial and temporal development of the cloud structures and patterns of solar surface irradiance transition seamlessly without any signs of a break. Moreover, ANAKLIM++ is capable of improving the forecast quality of NWP and nowcasting throughout the entire investigated forecast horizon by up to 37% regarding the RMSE.

**Author Contributions:** Conceptualization, I.U. and R.M.; methodology, I.U. and R.M.; software, I.U.; validation, I.U.; formal analysis, J.B. and R.M.; writing—original draft preparation, I.U.; writing—review and editing, R.M. and J.B.; visualization, I.U.; supervision, R.M. and J.B.; project administration, R.M.; All authors have read and agreed to the published version of the manuscript.

**Funding:** This research was funded by Gridcast, a project by the Federal Ministry for Economic Affairs and Energy (Bundesministerium für Wirtschaft und Energie, BMWi).

**Acknowledgments:** We want to thank Jörg Trentmann and Uwe Pfeifroth for providing the SARAH-2 data for our validation. Thanks to Michael Mott, Manuel Werner and Nils Rathmann for the introduction and support regarding the POLARA framework which was used for the nowcasting of SIS. POLARA was developed by the department of radar meteorology at the German Weather Service.

**Conflicts of Interest:** The authors declare no conflict of interest.

## Abbreviations

The following abbreviations are used in this manuscript:

| | |
|---|---|
| ANAKLIM++ | Adjustment of assimilation software for the reanalysis of climate data |
| BMWi | Bundesministerium für Wirtschaft und Energie (Federal Ministry for Economic Affairs and Energy) |
| CM SAF | Climate Monitoring Satellite Application Facility |
| COSMO | Consortium for Small Scale Modeling |
| DNI | Direct Normal Irradiance |
| DWD | Deutscher Wetterdienst (German Weather Service) |
| ECMWF | European Centre for Medium-Range Weather Forecasts |
| EU | Europe |
| GHI | Global Horizontal Irradiance |
| ICON | Icosahedral Non-Hydrostatic Model |
| IFS | Integrated Forecasting System |
| MAE | Mean Absolute Error |
| MSG | Meteosat Second Generation |
| NDFD | National Digital Forecast Database |
| NN | Neural Networks |
| NWC | Nowcasting |
| NWP | Numerical Weather Prediction |
| OpenCV | Open Source Computer Vision |
| RMSE | Root Mean Square Error |
| PV | Photovoltaic |
| SARAH-2 | Surface Radiation Data Set–Heliosat |
| SESORA | Seamless Solar Radiation |
| SIS | Surface Incoming Shortwave Radiation |
| SPECMAGIC NOW | Spectrally Resolved Mesoscale Atmospheric Global Irradiance Code for Nowcasting |
| TSO | Transmission System Operator |
| WRF | Weather Research and Forecasting |

## Appendix A. Investigated Cases

**Table A1.** List of investigated cases with corresponding main weather situation and cloud type over Germany. The upper row shows the MAE and the lower row shows the RMSE, respectively. The unit for the error measures is W/m$^2$.

| Date | Weather Situation/ Cloud Type | Day Time (UTC) | | | | |
|---|---|---|---|---|---|---|
| | | 10 | 11 | 12 | 13 | 14 |
| 7 August 2017 | high pressure cirrus | 48.30 68.14 | 61.64 91.35 | 67.92 101.81 | 70.94 101.65 | 67.22 91.38 |
| 11 August 2017 | stratiform precipitation stratus | 49.12 72.66 | 60.80 91.68 | 63.21 94.60 | 62.88 90.85 | 59.19 81.76 |
| 15 August 2017 | convection cumulus nimbus | 47.19 70.99 | 61.08 94.97 | 68.48 104.18 | 73.96 107.58 | 69.32 96.68 |
| 28 August 2017 | high pressure cirrus | 51.01 70.78 | 67.50 96.63 | 73.41 106.4 | 71.34 101.67 | 65.25 89.50 |
| 29 August 2017 | high pressure cirrus | 48.05 70.22 | 61.97 92.45 | 66.57 99.91 | 67.04 98.03 | 61.18 86.24 |
| 1 September 2017 | stratiform precipitation stratus | 50.61 73.75 | 67.90 100.72 | 73.91 106.94 | 73.13 102.02 | 64.57 86.28 |
| 7 September 2017 | broken clouds cumulus | 52.58 72.75 | 70.20 95.96 | 75.34 100.94 | 70.30 92.28 | 59.89 75.54 |
| 17 September 2017 | broken clouds cumulus | 48.19 71.23 | 67.19 99.70 | 72.16 104.32 | 66.35 91.82 | 53.82 70.97 |
| 26 September 2017 | convection cumulus nimbus | 51.18 68.50 | 66.56 90.47 | 71.07 95.84 | 65.65 86.40 | 52.12 66.54 |
| 30 September 2017 | front & convection stratus & cumulus | 41.83 59.28 | 54.12 77.83 | 56.35 79.33 | 51.44 69.97 | 41.38 54.94 |
| 1 October 2017 | front & convection stratus & cumulus | 56.31 70.67 | 66.78 85.74 | 67.82 86.00 | 62.36 76.57 | 53.00 62.67 |
| 2 October 2017 | stratiform precipitation stratus | 60.52 76.69 | 73.19 95.71 | 74.20 97.24 | 67.50 87.35 | 54.72 69.68 |
| 3 October 2017 | broken clouds cumulus | 53.53 68.14 | 60.70 79.91 | 60.93 79.89 | 55.04 69.93 | 45.34 55.97 |
| 4 October 2017 | stratiform precipitation stratus | 51.49 65.95 | 58.76 77.37 | 58.12 76.85 | 52.19 68.89 | 41.70 55.14 |
| 7 October 2017 | stratiform precipitation stratus | 47.81 63.50 | 51.14 72.03 | 49.07 68.84 | 41.82 56.10 | 33.14 41.51 |

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
