# Peer review of "Development of a Seamless Forecast for Solar Radiation Using ANAKLIM++"

_remotesensing, doi:10.3390/rs12213672_

Round 1

Reviewer 1 Report

Dear Authors,

thank you for this contribution. I have only some minor remarks.

L37/38 SIS is called "solar surface irradiance" througout the text, At first I wondered why you did not use SSI then? until I found
in the abbreviations L381 the definition "surface incoming shortwave radiation".
I would prefer surface solar irradiance as the solar surface is far away... please check.

L381 Abbreviations: SAHRAH-2 is a Surface Radiation Dataset - Heliosat NOT Solar Surface Radiation Dataset - Heliosat, for the same reason.

L120/121 check sentence: used is used twice...

L127  Minus sign - after Mathiesen et al.

L242/243 check sentence: based on for -> either based on or for

L245 persistence: Here it is important to mention, that naiv persistence is used, (if done so).
If you would have used the cloud index / clearsky index persistence, instead, the SIS would better follow the daily pattern.
In clear or overcast days this would not cause a big error. Or if you used cloud index persistence, that should be mentioned. 

L298-305 whole paragraph should go to the introduction.

L319/320 check sentence: is that is

In Fig.4 (d)

L37/38 SIS is called "solar surface irradiance" througout the text, At first I wondered why you did not use SSI then? until I found
in the abbreviations L381 the definition "surface incoming shortwave radiation".
I would prefer surface solar irradiance as the solar surface is far away... please check.

L381 Abbreviations: SAHRAH-2 is a Surface Radiation Dataset - Heliosat NOT Solar Surface Radiation Dataset - Heliosat, for the same reason.

L120/121 check sentence: used is used twice...

L127  Minus sign - after Mathiesen et al.

L242/243 check sentence: based on for -> either based on or for

L245 persistence: Here it is important to mention, that naiv persistence is used, (if done so).
If you would have used the cloud index / clearsky index persistence, instead, the SIS would better follow the daily pattern.
In clear or overcast days this would not cause a big error. Or if you used cloud index persistence, that should be mentioned. 

L298-305 whole paragraph should go to the introduction.

L319/320 check sentence: is that is

(In Fig.4(d): please show the points of your analysis on the lines, as a line suggests an analytic function.) 

Author Response

Please find our response in the attached file.

Reviewer 2 Report

Review of remotesensing-973025:

Development of a seamless forecast for solar surface irradiance using ANAKLIM++

by

Isabel Urbich, Jörg Bendix , Richard Müller

GENERAL COMMENTS: The paper analyzes the novel approach for a blending between nowcasting and numerical weather prediction (NWP) for the solar surface irradiance (SIS) for a forecast horizon of 1–5h examining 15 cases with different weather situations.

The article is well structured, clear and very interesting. In my opinion it can be published after a minor revision.

SPECIFIC COMMENTS:

  1. 1, L. 25-29/P.2, L. 61-67: In the introduction when the NWP is presented for the forecast of surface solar irradiance it has not been analyzed the impact of data assimilation on the NWP. For example, this recent paper (Gentile et al, https://doi.org/10.3390/rs12060920 ) shows that the assimilation of SEVIRI radiance improves the performance of WRF especially in the first 3-hour forecast. The same interval where, usually, the nowcasting outperforms the NWP. These results can be added into the introduction and discussed in the Summary and Conclusions.

  1. 3, L. 105-109: What is the icon model initialized with?

  1. 3, L. 118: Use the acronym SIS, check this error for all the text

  1. 4, L. 129-130: This statement is not true: a higher horizontal resolution allows a more correct reproduction of the explicit convention and therefore of the cloud cover and consequently of the surface solar irradiance. Rewrite the sentence.

Arbizu-Barren, C.; Pozo-Vázquez, D.; Ruiz-Arias, J.A.; Tovar-Pescador, J. Macroscopic cloud properties in the WRF NWP model: An assessment using sky camera and ceilometer data. J. Geophys. Res. Atmos. 2015, 120, 10297–10312.

  1. 4, L. 149-154. Which is the horizontal resolution of the data?

  1. 6, Table 2: Why the 2 NWP have 2 different weights? Due to their horizontal resolutions? Explain in the text.

  1. 9, L. 260-262: For completeness the normalized scores could be evaluated to reduce the dependence of the statistical indexes on the values of solar irradiance.

Author Response

(The authors gave the same response as above.)

Reviewer 3 Report

Please, look at the attached file.

Author Response

(The authors gave the same response as above.)

Round 2

Reviewer 3 Report

Article Title: Development of a seamless forecast for solar surface irradiance using ANAKLIM++

Brief summary:

In this article, to provide the short-term forecast of solar surface irradiance, the authors suggested the blending method in which three types of input data (i.e., satellite-based nowcasting and the NWP forecast from ICON of DWD and IFS of ECMWF) are blended using ANAKLIM++ software. Though the aim of ANAKLIM++ algorithm is to assimilate the 2-D observation data with the forecast from the NWP system, in this study, this algorithm is used to blend various input data to provide more reliable forecast of solar surface irradiance for 1-5h forecast hours. For the verification with the SARAH-2 data as a reference, it is shown that the solar irradiance forecast from this used algorithm is much better than other data products (e.g., NWP forecasts from ICON and IFS, nowcasting), thus reflecting the reliability of this algorithm.

Thank you for your dedicated reply about my comments at 1st round. The recent version of this article is now well organized and eligible to be published in the Remote Sensing Journal. However, to enhance the comprehension of the reader, please add extra one sub-figure in Figure 4.

Minor comments

  1. Please, add the standard deviation (STDDEV) results in Figure 4

As you mentioned in 1st round revision, the significant positive/negative biases of the ICON forecasts and nowcasting are well compensated by blending two inputs as well the IFS forecasts. Thus, the forecast derived from ANAKLIM++ algorithm has quite small biases that are close to zero. The results of RMSEs also show the significant improvement for the ANAKLIM forecast as compared with other input data. However, as you know, since the RMSE is an error value that is mixed with both the standard deviation and the mean bias errors, it is difficult to assess the forecast performance in terms of the forecast spatial variability. In addition, in this case, as the mean bias is so large, the RMSEs seem not to be representative to show a different aspect of forecast error. Thus, to assess the forecast accuracy in terms of the forecast spatial variability, the standard deviation seems to be more informative to describe the overall forecast accuracy as compared with other input data.

Author Response

As was requested by reviewer 3 the authors added a subplot in Figure 4 (Figure 4(d)). This plot depicts the bias corrected RMSE since the reviewer suggested to reduce the influence of the bias from the RMSE to underline the spatial variability of the data. The added plot can confirm the hypothesis that the over- and underestimation of the individual input forecasts leads to a blended forecast which is able to outperform its input data. Since the input of the blendings was bias corrected the difference between the RMSE of the nowcasting and the merged forecasts decreases significantly, in comparison to Figure 4(b), where no bias correction was applied. However, the RMSE of the merged products is still lower than the one of nowcasting, because the error cancellation of large positive values versus large negative values within the merging process is reduced but not eliminated by the bias correction. As a consequence of this error cancellation the merged forecast products show the lowest RMSE. Furthermore, the respective errors of ICON, IFS and nowcasting are lower after a bias correction.